# Entity Factor: A Balanced Method for Table Filling in Joint Entity and Relation Extraction

**Zhifeng Liu ***[ID]**, Mingcheng Tao * and Conghua Zhou**

School of Computer Science and Communication Engineering, Jiangsu University, Zhenjiang 212013, China
* Correspondence: liuzf@ujs.edu.cn (Z.L.); 2222008057@stmail.ujs.edu.cn (M.T.)

**Abstract:** The knowledge graph is an effective tool for improving natural language processing, but manually annotating enormous amounts of knowledge is expensive. Academics have conducted research on entity and relation extraction techniques, among which, the end-to-end table-filling approach is a popular direction for achieving joint entity and relation extraction. However, once the table has been populated in a uniform label space, a large number of null labels are generated within the array, causing label-imbalance problems, which could result in a tendency of the model's encoder to predict null labels; that is, model generalization performance decreases. In this paper, we propose a method to mitigate non-essential null labels in matrices. This method utilizes a score matrix to calculate the count of non-entities and the percentage of non-essential null labels in the matrix, which is then projected by the power of natural constant to generate an entity-factor matrix. This is then incorporated into the scoring matrix. In the back-propagation process, the gradient of non-essential null-labeled cells in the entity factor layer is affected and shrinks, the amplitude of which is related to the size of the entity factor, thereby reducing the feature learning of the model for a large number of non-essential null labels. Experiments with two publicly available benchmark datasets show that the incorporation of entity factors significantly improved model performance, especially in the relation extraction task, by 1.5% in both cases.

**Keywords:** natural language processing; joint entity relation extraction; label imbalance

## 1. Introduction

Extracting specific entities and their relations from plain text is a fundamental task in natural language processing (NLP) and the basis of downstream tasks such as knowledge graph construction. The research topic of entity and relation extraction can be divided into two subtopics named entity recognition [1] and relation extraction [2]. The named entity recognition task aims to identify entities with specific meanings from plain text. The relation classification task aims to predict corresponding relational classes among the entities identified. The researchers further divide the extraction methods into pipeline extraction and joint extraction methods based on the sequence of the the two subtasks.

The traditional pipeline extraction approach is to first construct a model for extracting entities with specific meanings in the text [3], and later, another model to classify relations on the results of entities extracted by the previous named entity recognition model [4]. Although the pipeline extraction approach is easy to implement, the relational classification task inevitably suffers from error propagation from the named entity recognition task because the input value is the output value of the named entity recognition task, and researchers have been working on this problem for a long time. Recently, the joint extraction model [5–8] has become a popular method because of parameter sharing for entity recognition and relation classification. In the training process, the model can handle the error propagation of the entity recognition task internally, so as to avoid the error propagation problem in pipeline extraction.

End-to-end table filling is a common implementation of joint entity and relation extraction. Wang [9] strengthened the intrinsic connection between the two tasks in the same model by populating a table in a unified labeling space to achieve joint extraction of the entities and relations, but as shown in Figure 1, the table filled by the model has a serious category imbalance problem—the number of null labels in the table is much larger than in the numbers of other labels. This is a typical long-tailed label distribution problem, in which most samples are only a fraction of the label, which reduces the generalization of the model. Since the joint extraction of entity relationships in a unified label space with table filling is a relatively new approach, there is no very suitable method for too many null labels in a table, because the reason for its appearance is generated by the characteristics of the table, and historical label-balancing methods have limitations in this scenario. When some downstream work of NLP, such as building knowledge graphs, requires entity relationship extraction, if it uses table filling, it can consider incorporating entity factors to increase the extraction performance of its model.

|  | Dole | 's | wife | , | Elizabeth | , | is | a | native | of | Salisbury | , | N.C |
|---|---|---|---|---|---|---|---|---|---|---|---|---|---|
| Dole | Peop | ⊥ | ⊥ | ⊥ | ⊥ | ⊥ | ⊥ | ⊥ | ⊥ | ⊥ | ⊥ | ⊥ | ⊥ |
| 's | ⊥ | ⊥ | ⊥ | ⊥ | ⊥ | ⊥ | ⊥ | ⊥ | ⊥ | ⊥ | ⊥ | ⊥ | ⊥ |
| wife | ⊥ | ⊥ | ⊥ | ⊥ | ⊥ | ⊥ | ⊥ | ⊥ | ⊥ | ⊥ | ⊥ | ⊥ | ⊥ |
| , | ⊥ | ⊥ | ⊥ | ⊥ | ⊥ | ⊥ | ⊥ | ⊥ | ⊥ | ⊥ | ⊥ | ⊥ | ⊥ |
| Elizabeth | ⊥ | ⊥ | ⊥ | ⊥ | Peop | ⊥ | ⊥ | ⊥ | ⊥ | ⊥ | Live-In | Live-In | Live-In |
| , | ⊥ | ⊥ | ⊥ | ⊥ | ⊥ | ⊥ | ⊥ | ⊥ | ⊥ | ⊥ | ⊥ | ⊥ | ⊥ |
| is | ⊥ | ⊥ | ⊥ | ⊥ | ⊥ | ⊥ | ⊥ | ⊥ | ⊥ | ⊥ | ⊥ | ⊥ | ⊥ |
| a | ⊥ | ⊥ | ⊥ | ⊥ | ⊥ | ⊥ | ⊥ | ⊥ | ⊥ | ⊥ | ⊥ | ⊥ | ⊥ |
| native | ⊥ | ⊥ | ⊥ | ⊥ | ⊥ | ⊥ | ⊥ | ⊥ | ⊥ | ⊥ | ⊥ | ⊥ | ⊥ |
| of | ⊥ | ⊥ | ⊥ | ⊥ | ⊥ | ⊥ | ⊥ | ⊥ | ⊥ | ⊥ | ⊥ | ⊥ | ⊥ |
| Salisbury | ⊥ | ⊥ | ⊥ | ⊥ | ⊥ | ⊥ | ⊥ | ⊥ | ⊥ | ⊥ | Loc | Loc | Loc |
| , | ⊥ | ⊥ | ⊥ | ⊥ | ⊥ | ⊥ | ⊥ | ⊥ | ⊥ | ⊥ | Loc | Loc | Loc |
| N.C | ⊥ | ⊥ | ⊥ | ⊥ | ⊥ | ⊥ | ⊥ | ⊥ | ⊥ | ⊥ | Loc | Loc | Loc |

**Figure 1.** Table for joint entity and relation extraction. Each cell in the table corresponds to a word pair, the square part on the main diagonal is the entity, and the rectangular part on the off-diagonal is the relation. Any label is produced by averaging the pooling of the encoder output.

In this paper, we reexamine the concrete representation of the problem in the table of the population-based joint entity and relation extraction method. The joint entity and relation extractor takes two kinds of actions, coding and decoding, and there is the problem of the category imbalance problem in the the encoding filling phase. Figure 1 shows the result of decoding the table after filling. As can be seen in the figure, the entity is decoded in the main diagonal part, where the purple part indicates that the cell is a null label of entity, and since the relation depends on the entity, the cell of its corresponding row and column, i.e., the gray part, will not decode any relationship either. We call this part of the cell a non-essential null label cell, and it is obvious that this part of the non-essential null label cell is the majority of the table. Figure 2 shows the statistics of the two datasets used in this paper regarding non-essential null labels. As can be seen in the line graphs, the percentage of non-essential null labels increases in proportion with sentence length, and there is a dramatic increase between 0 and 10, rising rapidly to over 80%. In addition, the histogram shows that the vast majority of sentences' length in both datasets are longer

than 10, which shows the overrepresentation of non-essential null tags in almost the entire dataset of text. Intuitively, how to mitigate the negative effects of such non-essential null labels on the model is the key at hand.

This study focuses on the label imbalance of the joint entity and relation extraction based on table filling, which reduces the feature learning of non-essential null labels in the table by incorporating entity factors and improves the generalization ability of the model. In explaining these research results, they can be interpreted in terms of the back-propagation process of model training; the gradient of non-essential null-labeled units shrinks after incorporating entity factors, and subsequently, the model reduces the feature learning of such null-labeled cells. In summary, the main contribution of this paper is to propose a balancing method for entity factors that supports Softmax cross-entropy continuity while alleviating the label-imbalance problem in filling tables in joint entity and relationship extraction and improving the generalization ability of the schema.

The rest of this paper is organized as follows. Section 2 describes the related work. Section 3 parses out the structure of the model from the encoder–decoder perspective, respectively. Section 4 compares the performance of the current approach with those of other models. Section 5 summarizes the conclusions.

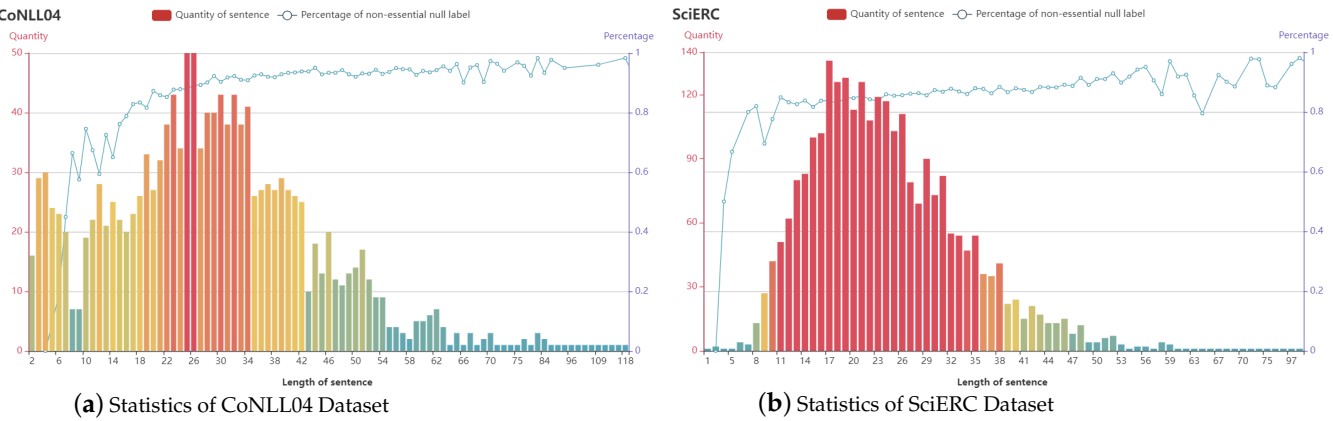

(**a**) Statistics of CoNLL04 Dataset          (**b**) Statistics of SciERC Dataset

**Figure 2.** In the datasets CoNLL04 and SciERC, the percentage of non-essential null labels is positively correlated with the sentence length, and the percentage of non-essential null labels is above 80% for most of the data.

## 2. Related Work

For entities and relation extraction, researchers propose several approaches to achieve this goal. The pipeline extraction method [10] neglects the connection between the two tasks and has error propagation problems. To solve this problem, researchers proposed an joint entity and relation extraction approach that exploits the interrelationships between the named entity recognition task and the relation classification task to mitigate the error propagation problem [11] by transforming the extraction of entity relations into a table filling problem. The entries in the i-th row and j-th column of the table correspond to the i-th and j-th words in the word-pairs input sentence, and the main diagonal entries in the table are entity labels. The remaining labels are relation labels. Currently, the table-filling method is one of the mainstream joint entity and relation extraction methods [12]. Although the entity and relation models in these joint extraction models share a set of encoders, they have their own independent set of label spaces, whereby Wang [9] proposed a unified space-based joint entity and relation extraction model and optimized multiple public datasets. However, it still suffers from a label imbalance.

Currently, the suggested solutions for the long-tail problem of labels fall into three general groups: Solutions for the input values of the model, such as oversampling or downsampling [13–15]. Solutions for the output values of the model, such as post hoc correction of decision thresholds [16,17] and loss weighting [18,19]. Solutions modifying the internal structure of the model, e.g., modifying the loss function [20–22]. However, these solutions do not adequately address the label-imbalance problem in the current models. Downsampling, for example, reduces the number of majority class labels by random discarding input corpus text, but the category imbalance problem in the filling out form is an internal problem that exists in almost every text. Moreover, the loss correction approach sacrifices the consistency of the softmax cross-entropy [23]. Therefore, the existing technique cannot be an optimal choice in the current environment.

## 3. Methodology

Our model is based on the UNIRE model proposed by Wang et al. [9]. The whole model is divided into two parts: encoder and decoder. This section introduces the structure of the whole model in detail. Figure 3 shows an overview of the model's architecture.

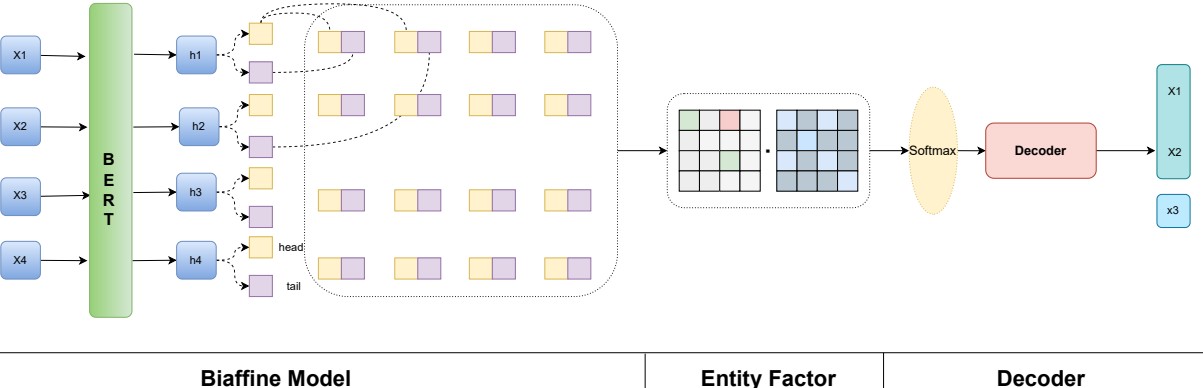

**Figure 3.** Architecture diagram of the model. After the initial score matrix is given by the biaffine model, the final score matrix is acquired by incorporating the entity factors, and the decoder decodes the entity and relation labels depending on the final score.

### 3.1. Problem Definition

Given a sentence input $s = x_1, x_2, \ldots, x_{|s|}$ ($x_i$ is a word), extract a set of entities $\mathscr{L}_e$ and relations $\mathscr{L}_r$ that exist in the sentence. Entity $e$ is the span of a continuous word sequence with a predefined entity type $e.type \in \mathscr{E}$. Relation $r$ is a predefined relation type $r.type \in \mathscr{R}$ that exists in a triplet $(e_1, e_2, r)$, where $e_1$ and $e_2$ are entities. $\mathscr{E}$ and $\mathscr{R}$ represent, respectively, predefined sets of entity and relational types—that is, the labeling space for the entire model; i.e., $\mathscr{L} = \mathscr{E} \cup \mathscr{R} \cup null$. For example, as shown in Figure 1, predefined types are entities $\mathscr{E} = \{Loc, Org, Peop, Other\}$ and relations $\mathscr{R} = \{Work\_for, Kill, OrgBased\_In, Live\_in, Located\_In\}$; moreover, entities $e_1 = (Dole; Peop)$, $e_2 = (Elizabeth; Peop)$, $e_3 = (Salisbury, N.C; Loc)$, and relation $r_1 = (Elizabeth, Salisbury, N.C, Live\_in)$ can be parsed from the sentence "Dole's wife, Elizabeth, is a native of Salisbury, N.C.".

### 3.2. Encoder

For an input sentence, we use a pre-trained language model (BERT model, etc.) to obtain the contextual representation of each word in the sentence:

$$h_1, h_2, \ldots, h_n = BERT(x_1, x_2, \ldots, x_n), \tag{1}$$

where $x_i$ is the i-th word in the sentence, $h_i$ is the contextual representation of word $i$, and $h_i \in \mathbb{R}_d$. Then, we project $h_i$ into the roles of head and tail with two reduced-dimension multilayer perceptrons (MLPs):

$$h_i^{head} = MLP_{head}(h_i), h_i^{tail} = MLP_{tail}(h_i), \tag{2}$$

where $h_i^{head} \in \mathbb{R}^d$, $h_i^{tail} \in \mathbb{R}^d$. Afterwards, the initial label score of the word pair $g_{i,j}$ is calculated with a deep biaffine attention model [24]:

$$g_{i,j} = Biaff(h_i^{head}, h_j^{tail}), \tag{3}$$

where $g_{i,j} \in \mathbb{R}^{|\mathcal{L}|}$. Given that relations dependent on entities exist, the non-essential null label will be obtained based on the initial label scores $g_{i,j}$. When a cell has the highest null label score, word $i$ can be considered not an entity temporarily; therefore, the word pairs in its row and column will not have any relation labels. These cell is defined as a non-essential null label, from which can produce an entity-factor matrix to alleviate the adverse effects of these non-essential null labels on the model's performance. When generating the entity-factor matrix, different entity factors shall be produced by the initial label scores. The formula is

$$w_{i,j} = \begin{cases} 1, P(g_{i,i}) \in \mathcal{L}_e \text{ or } P(g_{j,j}) \in \mathcal{L}_e \text{ or } i = j, \\ e^{\frac{n}{|s| \times |s|}}, P(g_{i,j}) \in null \text{ or } P(g_{j,i}) \in null. \end{cases} \tag{4}$$

where $n$ is the number of non-essential null labels for the current input sentence, $n = 2 \cdot \sum_{|s|-m}^{|s|-1} i$, and $m$ is the number of non-entity label words. Afterwards, integrate the entity factor into the table (Figure 3); the final word pair $(x_i, x_j)$ has a label score $g_{i,j}' = w_{i,j} \cdot g_{i,j}$. After yielding the score vector, feed it into the softmax function to obtain the corresponding labels, generating a probability distribution over the label space:

$$P(y_{i,j}|s) = Softmax(dropout(g_{i,j}')), \tag{5}$$

The encoder model has a loss function:

$$Loss = -\frac{1}{|s|^2} \sum_{i=1}^{|s|} \sum_{j=1}^{|s|} logP(y_{i,j} = y_{i,j}'|s), \tag{6}$$

where $y_{i,j}'$ is a gold label. The entity-factor matrix is incorporated prior to normalization, and only the non-essential null-labeled cells correspond to entity factors that are not one. In the training stage of the model, the loss occurs at the entity-factor layer, $loss = P(y_{i,j}|s) - y_{i,j}'$, after which the loss will be affected by the entity factor. If the probability of each label varies somewhat, it shrinks before spreading backward to the next layer, so the entity factors will start to be incorporated after a period of model training as a way to mitigate non-essential null label features that the encoder learns too much about. Entity factors support the softmax cross-entropy consistency while mitigating the negative effects of non-essential null labels on encoders. The encoder is as Algorithm 1.

### 3.3. Decoder

This part follows the view of wang [9] in that the decoding process is divided into three parts: span decoding, entity type decoding, and relation type decoding.

For a given sentence, compute the Euclidean distance between two adjacent rows or columns from the row and column perspectives in its probability tensor $P \in \mathbb{R}^{|s| \times |s| \times \mathcal{L}}$, respectively, when the average of these two distances is greater than a threshold, which is considered here to be a demarcation point. The sequence between two demarcation points is considered as a span.

For any span $(i, j)$, the average score $t' = argmax_{t \in \mathcal{L}_e \cup null} Avg(P_{i:j,i:j,t})$ of the square area in the table; if $t' \in \mathcal{L}_e$, the span is considered an entity; otherwise, it is not an entity.

For any entity pairs $(e_1, e_2)$, their spans are $(i, j)$ and $(m, n)$, respectively, and the average score of the rectangular region corresponding to the two spans in the label score

table is $r' = argmax_{r \in \mathscr{L}_r \cup null} Avg(P_{i:j,m:n,r})$. If $r' \in \mathscr{L}_r$, the relation lies on the entity pair; otherwise, no relation exists. The decoder is as Algorithm 2.

---

**Algorithm 1:** Encoder

---

**Input:** sentence $s = x_1, x_2, ..., x_{|s|}$ ($x_i$ is a word)
**Output:** categorical probability distribution table $P$
  **for** $x_i$ in $s$ **do**
    $h_i = BERT(x_1, x_2, ..., x_{|s|})$
  **end for**
  **for all** $h_i$ **do**
    $h_i^{head} = MLP_{head}(h_i)$
    $h_i^{tail} = MLP_{tail}(h_i)$
  **end for**
  **for all** $(h_i^{head}, h_j^{tail})$ **do**
    $g_{i,j} = Biaff(h_i^{head}, h_j^{tail})$
  **end for**
  set $w[][] = \{1\}, w \in \mathbb{R}^{|\mathfrak{z}| \times |\mathfrak{z}|}$
  **for all** $g_{i,j}$ **do**
    **if** $i \neq j$ and $max(g_{i,i})$ is $g_{i,i}[0]$ or $max(g_{j,j})$ is $g_{j,j}[0]$ **then**
      $w_{i,j} = e^{\frac{2 \cdot t}{|s| \times |s|}}$, where $t = \sum_{k=|s|-m}^{|s|-1} k$
    **end if**
    $g_{i,j}' = g_{i,j} \cdot w_{i,j}$
  **end for**
  $P = Softmax(g')$
  **return** $P$

---

**Algorithm 2:** Decoder

---

**Input:** categorical probability distribution table $P$
**Output:** span, entity and relation list
  **for** $i$ of row and column in $P$ **do**
    $l_{row} = l_2(p_{i-1}^{row}, p_i^{row})$
    $l_{col} = l_2(p_{i-1}^{col}, p_i^{col})$
    **if** avg($l_{row}, l_{col}$)<$\alpha$ **then**
      span_list.add($i$)
    **end if**
  **end for**
  **for** span($i,j$) in span_list **do**
    **if** ent = $argmax_{t \in \mathscr{L}_e \cup null} Avg(P_{i:j,i:j,t}) \in \mathscr{L}_e$ **then**
      entity_list.add(span($i,j$),ent)
    **end if**
  **end for**
  **for** span($i,j$), span($m,n$) in entity_list **do**
    **if** rel = $argmax_{r \in \mathscr{L}_r \cup null} Avg(P_{i:j,m:n,t}) \in \mathscr{L}_r$ **then**
      rel_list.add(span($i,j$), span($m,n$), rel)
    **end if**
  **end for**
  **return** span_list, entity_list, rel_list

---

## 4. Experiment and Results

This section evaluates the effectiveness of the entity-factor method for table filling in two publicly available datasets, ConLL04 and SciERC.

*4.1. Dataset*

Two publicly available entity relation datasets, ConLL04 [25] and SciERC [26], were experimented with. Table 1 shows the statistics of these two datasets. Figure 1 shows the non-essential null labels in the two datasets. It is evident that in most sentences in both datasets, the percentage of non-essential null labels is above 80%, and how severely the model is affected by non-essential null labels.

**Table 1.** Statistics of the datasets.

| Dataset | #sents | #ents(#types) | #rels(#types) |
|---|---|---|---|
| ConLL04 | 1441 | 5349(4) | 2048(5) |
| SciERC | 2687 | 8094(6) | 5463(7) |

*4.2. Evaluation*

Following the suggestion of Yi [27], accuracy (P), recall (R), and F1 score were used as evaluation criteria. In addition, a strict evaluation criterion was applied; i.e., a predicted entity is considered correct when it has the right type and boundaries. A predicted relation is considered correct when the predicted relation type and the two entities on which it depends on are correct.

*4.3. Implementation Details*

To verify the validity of entity factors in a table-filling approach, we will compare the following models in two different pre-trained language models, bert-base-uncased [28] and scibert-scivocab-uncased [29].

PURE: This model uses a pipeline approach to implement the task of extracting entities and relationships, and the model hyperparameters follow the values recommended in its paper.

UNIRE: This model is our base model, which uses joint entity and relation extraction to extract entities and relations, and the model's hyperparameters follow the values recommended in its paper.

Logit Adjustment: The model uses UNIRE as the base model and Logit adjustment as the treatment of label imbalance.

Entity Factor: The model uses UNIRE as the base model and entity factors as the way to handle label imbalance.

All experiments were conducted in an Intel(R) Core i7-10700 CPU and NVDIA 3080 GPU environment, where the hyperparameters of the Logit adjustment and entity-factor models used the values of the base model.

*4.4. Performance Comparison*

Table 2 summarizes the performances of all experimental models on both public datasets. Performance data for the PURE [30] model on the SciERC dataset are from the original literature. Figures 4 and 5 show the training performances of the three models, UNIRE, a joint entity relationship extraction model based on table filling, and the models incorporating entity factors and logit adjustment on the basis of this model. It can be seen that the model with the logit adjustment converged faster, but the performance in the subsequent process was comparable to that of UNIRE, and the UNIRE model incorporating entity factors performed comparably to the UNIRE model initially, but surpassed UNIRE in the later stages of training, especially in relation extraction. In the dataset ConLL04, our model performed as well as UNIRE on entity recognition task, but scored highest in the relation classification task, leading by more than 1.5 percentage points in F1 scores. In the SciERC dataset, UNIRE outperformed PURE in the entity recognition task but lagged much behind in the relationship extraction task. Both label balancing methods improved UNIRE's relation extraction. The incorporation of the entity factor resulted in a more significant improvement: as much as 4.1% improvement in the F1 score for the relationship extraction

task, 1.5% more than the second-place method; and the highest score achieved for entity recognition, at 0.8% higher than the second-place method.

In general, our model achieves very competitive performance on both CoNLL04 and SciERC. PURE adopts pipeline extraction, and although it performs well in the entity recognition task, there is still error propagation in the relation classification task, so it is not better than the joint extraction model. UNIRE employs table filling to perform the task of entity relation extraction, but after table filling, non-essential null labels are often much larger in the table than other labels, challenging the generalizability of the model. The logit adjustment approach proposed by Aditya is a relatively advanced way to deal with the long-tail problem, but it clearly does not play much of a role in the table-filling task. Our model is based on the table-filling model UNIRE and incorporates entity factors. As seen in Figures 4 and 5 and Table 2, the UNIRE model with the incorporation of entity factors outperforms UNIRE, and the entity factor approach is more applicable to this form of table filling compared to the balanced approach of logit adjustment, which mitigates the adverse effects of non-essential null labels on the model and improves the extraction performance of the model. Since the non-essential null labels are only present in the off-diagonal region of the table, the model performs better in the relation classification task than in the entity recognition task. The results of this experiment also confirm the validity of our proposed idea of adding entity factors to the table-filling method.

**Table 2.** Experimental performance of the models on two datasets.

| Dataset | Model | Encoder | Entity | | | Relation | | |
|---|---|---|---|---|---|---|---|---|
| | | | P | R | F1 | P | R | F1 |
| CoNLL04 | PURE [30] | $BERT_{BASE}$ | - | - | **88.1** | - | - | 68.4 |
| | UNIRE [9] | $BERT_{BASE}$ | 87.6 | 88.5 | **88.1** | 68.3 | 71.1 | 69.7 |
| | Logit-Adjust [23] | $BERT_{BASE}$ | 86.9 | 88.2 | 87.6 | 69.7 | 68.7 | 69.2 |
| | ours | $BERT_{BASE}$ | 87.7 | 89.1 | 88.0 | 69.8 | 72.7 | **71.2** |
| SciERC | PURE [30] | SciBERT | - | - | 68.2 | - | - | 36.7 |
| | UNIRE [9] | SciBERT | 67.1 | 70.6 | 68.8 | 34.8 | 34.1 | 34.4 |
| | Logit-Adjust [23] | SciBERT | 65.6 | 70.8 | 68.1 | 34.1 | 43.0 | 38.0 |
| | ours | SciBERT | 67.1 | 72.4 | **69.6** | 39.7 | 39.3 | **39.5** |

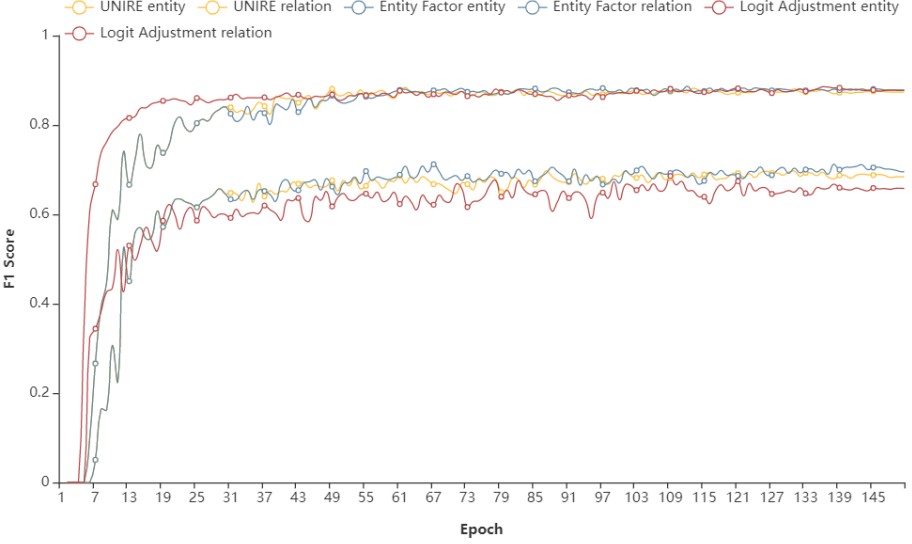

**Figure 4.** Training performance of models on the CoNLL04 dataset.

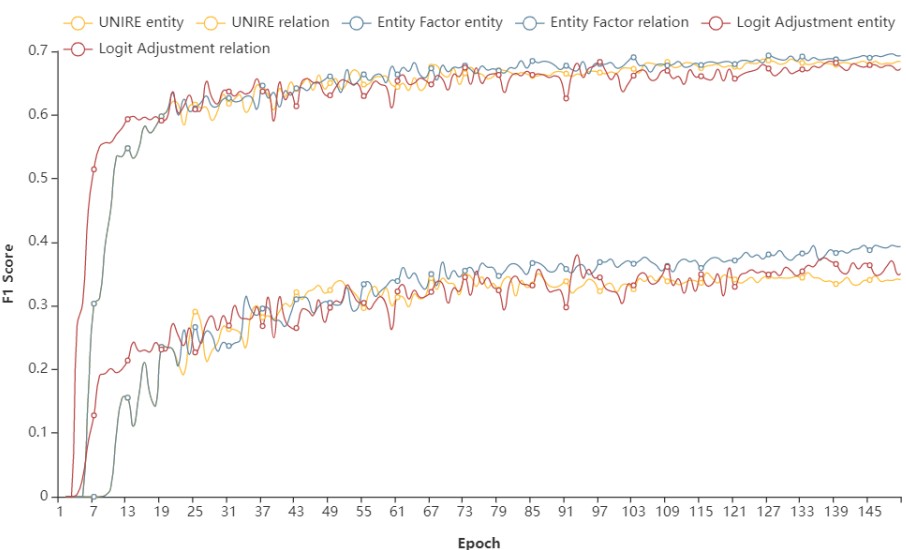

**Figure 5.** Training performance of models on the SciREC dataset

## 5. Conclusions

In this study, we performed joint entity and relation extraction in a table-filling manner, and we proposed a simple but effective way to alleviate the label-imbalance problem caused by too many null labels in the table. In model training, the method generates an entity factor based on the percentage of null labels in the table after the table is filled. Then, it incorporates the entity factor into all non-essential null label units in the table, which will shrink the gradient of such null-label units in the model via back-propagation while supporting softmax cross-entropy continuity, reducing the model's feature learning for massive null labels. Experiments on both datasets showed that the model achieves better performance in the entity and relation extraction tasks after incorporating the entity factor.

**Author Contributions:** Conceptualization, Z.L. and M.T.; methodology, M.T.; formal analysis, M.T.; investigation, M.T.; writing—original draft preparation, M.T.; writing—review and editing, C.Z. and Z.L.; supervision, Z.L. and C.Z. All authors have read and agreed to the published version of the manuscript.

**Funding:** This research received no external funding.

**Conflicts of Interest:** The authors declare no conflict of interest.

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
