# Peer review of "Entity Factor: A Balanced Method for Table Filling in Joint Entity and Relation Extraction"

_electronics, doi:10.3390/electronics12010121_

Round 1
Reviewer 1 Report
Dear sir
I have no comment about this paper; except to check grammar and linguistic mistakes
Good Luck
Author Response
Thank you very much for your comments. We have improved the English, grammar, and tense consistency in the revised manuscript.
Reviewer 2 Report
The following are the observations:
1. The Conclusion section is weak. It needs to be revised thoroughly in terms of results achieved and authors' observations. The claim "effective way to improve the performance of entity and relation joint extraction based on a table-filling approach." need to be justified by providing evidence of experimental results.
2. Figures are not discussed thoroughly. Especially the results in Figure 4 and 5 are not discussed at all.
Overall, the paper needs thorough revision and cannot be accepted in its present form.
Author Response
Thanks a lot for your comments. About Our respones, please see the attachment. We have rewritten some contents in revised manuscript.

Reviewer 3 Report
1) In my opinion, the topic of the article ranks in the basic sciences, but this does not exempt the authors from presenting practical implications. In its current form, this is an article "about algorithms, for algorithms". What results from this research for the practice and development of innovation? How should business interpret the results of these studies? Please comment.
2) In my opinion, the article has a relatively good structure. In the introduction, I suggest highlighting the research gap. It is also good practice to include a concise presentation of the structure of the entire article at the end of the first section 1. Introduction.
3) In my opinion, the article is technical and engineering in nature. What scientific problem did the study authors solve? What's new in this research? What new approach did the authors use? Please comment.
4) The article lacks a solid discussion of the possibilities of practical application of "Balanced Method for Table Filling in Joint Entity Relation Extraction". Please comment.
5) In the first section of the article, line 52, the authors referred to figures (page 2), but these figures are only on pages 6-7. According to the MDPI guidelines: All Figures, Schemes and Tables should be inserted into the main text close to their first citation and must be numbered following their number of appearance (Figure 1, Scheme I, Figure 2, Scheme II, Table 1, etc.) .
Author Response

(The authors gave the same response as above.)

Round 2
Reviewer 2 Report
The Conclusion section and description of the figures are revised. However, the overall read of the paper still needs to be maintained from the readers' perspective. The authors are advised to go through the paper once again and make appropriate changes to make it an informative and interesting read. The paper is now in good shape but minor language issues are to be addressed and readability needs to be improved.
Reviewer 3 Report
I thank the authors for improving the manuscript. The revised manuscript is satisfactory.